# SAFE: Spiking Neural Network based Audio Fidelity Evaluation

## Abstract

Recent advancements in generative AI have enabled the creation of highly realistic synthetic audio, posing significant challenges in voice authentication, media verification, and fraud detection. While deep learning models are frequently used for fake audio detection, they often struggle to generalize to unseen and complex manipulations, particularly partial fake audio, where real and synthetic segments are seamlessly combined. This paper explores the use of Spiking Neural Networks (SNNs) for fake and partial fake audio detection, an area that has not yet been investigated. SNNs, known for their energy-efficient computation and ability to process temporal data, offer a promising alternative to traditional Artificial Neural Networks (ANNs). We propose an SNN-based approach for fake audio detection and comprehensively evaluate its performance through a series of experiments, including hyperparameter tuning, cross-dataset generalization and partial fake audio detection. Our results show that SNNs achieve accuracy comparable to state-of-the-art ANN models with fewer number of parameters. Although, SNNs did not offer significant improvements in generalization capabilities, they provided advantages such as reduced model sizes and computational efficiency, making them more suitable for resource-constrained and real-time voice authentication applications. This study lays the groundwork for further exploration of SNNs in audio spoofing countermeasures, providing a foundation for future advancements in security-critical voice applications.

## 1 Introduction

Rapid advancements in generative AI have led to the creation of highly realistic synthetic media, including images, video, and audio. These technologies are proving to be highly useful across various domains such as entertainment, customer service, education, and healthcare (Ramdurai & Adhithya, 2023). However, the ease of generating convincing artificial content also introduces significant ethical, security, and societal challenges. While much attention has been focused on synthetic images and deepfake videos (Rana et al., 2022) synthetic speech and audio technologies are emerging as equally critical, particularly in areas where voice authenticity is essential, such as voice authentication systems, customer service, and media. Specifically, modern Text-to-Speech (TTS) (Shen et al., 2018; Dieleman et al., 2016; Ren et al., 2020) and Voice Conversion (VC) technologies (Kameoka et al., 2018; Qian et al., 2019; Hsu et al., 2016), which are driven by generative AI models, leverage advanced Deep Learning (DL) architectures to generate speech that is nearly indistinguishable from natural human voices (Mai et al., 2023; Prudký et al., 2023). The ability to produce such realistic synthetic speech has enabled malicious activities such as impersonation, fraud, and disinformation (Bleisch, 2024; Hickey, 2023). Moreover, one particularly concerning development is the seamless merging of synthetic and genuine audio segments, resulting in partial fake audio (Zhang et al., 2021) which further complicates the distinction between authentic and altered content.

The current approaches for detecting fake audio, or synthetic speech, initially relied on Machine Learning (ML) algorithms and have since evolved to incorporate advanced Deep Leaning (DL) models (Dixit et al., 2023). The performance of these models also relies heavily on the quality of the training datasets. Among various options, the `ASVSpoof-2019` (Wang et al., 2020b) and `Fake or Real` (Reimao & Tzerpos, 2019) datasets are widely used benchmarks for fake audio detection. Certain models proposed in the literature such as RawNet2 (Jung et al., 2020) and DeepSonar (Wang et al., 2020a) have shown considerable performance on these datasets because of their sophisti-

cated deep neural network (DNN) architectures over earlier traditional machine learning approaches (Singh & Singh, 2021; Rodríguez-Ortega et al., 2020). However, despite these advances, recent studies reveal that existing models continue to struggle with generalization to newer/previously unseen audio content generated by TTS and VC techniques (Chen et al., 2020). Although deep models are effective at learning complex patterns, they may overfit to specific datasets and fail to generalize well to unseen or evolving attack methods. In addition, the detection of partial fake audio—where real and synthetic audio segments are seamlessly merged—remains an underexplored area. This poses an additional challenge, as existing models typically assume fully fake or real audio, leaving them ill-equipped to handle such complex cases.

In light of these challenges, this paper explores the use of *Spiking Neural Networks (SNNs)* as a novel approach for detecting both fully fake and partial fake audio. While traditional Artificial Neural Networks (ANNs) have been extensively applied to this domain, SNNs have not yet been explored for fake audio detection. SNNs are inherently designed to process temporal data due to their ability to capture the timing and sequence of events, which makes them particularly suitable for tasks involving audio, which has rich temporal dynamics (Baek & Lee, 2024). By leveraging these capabilities, our work explores the feasibility of using SNNs for fake audio detection and comprehensively evaluate their performance across various tasks, including cross-dataset generalizability evaluation and the detection of partial fake audio. Our study aims to provide insights into how well SNNs may be suited for fake audio detection, offering lessons on their strengths and limitations in handling complex, evolving audio manipulation techniques. These insights can help guide future work on enhancing voice authentication, media verification, and fraud detection systems, where reliable and efficient detection of audio manipulations is increasingly important.

In this work, we propose two specific SNN models based on their suitability for handling temporal patterns in audio: a four-layer feed-forward SNN and a hybrid convolutional SNN. The feed-forward model is selected for its simplicity and efficiency in processing sequential data, making it an appropriate baseline for evaluating the core capabilities of SNNs in this context. The hybrid convolutional SNN, on the other hand, incorporates convolutional layers to capture more complex spatial-temporal features in the audio signal, making it more-suited for detecting intricate manipulations like partial fakes. Our result show that the proposed SNN and CSNN models performed comparably to ANN models when trained and tested on the same dataset, but struggled with cross-dataset generalization, similar to other baselines and prior works. Nevertheless, our CSNN model achieved 16.55% improvement over the state-of-the-art (SOTA) model (Firc et al., 2024) when trained on `Fake or Real` dataset and tested on `ASVspoof-2019` dataset. For the more challenging partial fake audio detection task, we achieve an accuracy of 85.59% on partial fake audio dataset created using `Fake or Real` dataset. To the best of our knowledge, this is the first work to apply SNNs to the task of fake audio detection, highlighting their potential in this rapidly evolving field. Our main contributions are as follows.

- **Propose two novel SNN models** for detecting both fake and partial fake audio, including a feed-forward SNN and a hybrid convolutional SNN model.
- **Conduct comprehensive hyperparameter tuning** experiments for SNN models, optimizing surrogate gradients (Fast Sigmoid, Arctangent), and loss functions (CE-count, CE-rate) to maximize performance in fake audio detection.
- **Perform a cross-dataset generalization study** using the `Fake or Real` and `ASVspoof-2019` datasets to assess robustness of SNNs in detecting fake audio across diverse and unseen data.
- **Develop and evaluate on a new partial fake audio dataset**, combining real and synthetic samples from the `Fake or Real` dataset, demonstrating SNNs' effectiveness in detecting partial fake audio at the frame level.

## 2 RELATED WORK

**Fake Audio Detection** Pipeline for fake audio detection typically involves feature extraction from input audio and classification. The traditional classification methods used handcrafted features such as MFCCs, CQCCs, and LPCCs, combined with machine learning classifiers such as Gaussian Mixture Models (GMMs) (Todisco et al., 2016). With the rise of deep learning, several models now employ end-to-end architectures that can learn features directly from raw audio, eliminating the

Table 1: A summary of related works.

| Model | Dataset | Accuracy (%) | EER (%) |
|---|---|---|---|
| Residual CNN (Alzantot et al., 2019) | ASVspoof | - | 6.02 |
| ASSERT (Lai et al., 2019) | | - | 6.70 |
| ResNet (Aravind et al., 2020) | | - | 5.32 |
| Siamese CNN (Lei et al., 2020) | | - | 8.72 |
| RawNet2 (Tak et al., 2021) | | - | 1.12 |
| Stacked TCN (Firc et al., 2024) | | - | 23.37 |
| `ASVspoof` Baseline1 (Wang et al., 2020b) | | - | 9.57 |
| `ASVspoof` Baseline2 (Wang et al., 2020b) | | - | 8.09 |
| CNN (Wijethunga et al., 2020) | FoR | 94.00 | - |
| VGG19 (Reimao & Tzerpos, 2021) | | 52.02 | - |
| TCN (Khochare et al., 2021) | | 92.00 | - |
| STN (Khochare et al., 2021) | | 80.00 | - |
| Stacked TCN (Firc et al., 2024) | | - | 6.99 |

need for explicit feature extraction. For instance, RawNet2 (Tak et al., 2021) leverages a deep convolutional neural network (CNN) to learn representations from audio data without requiring manual feature engineering. Recent deep learning models have achieved considerable success in detecting fake audio on benchmark datasets such as `ASVspoof-2019` and `Fake or Real` datasets. Table 1 summarizes the performance of these models in the research literature on both these benchmark datasets. Among these models, only models that have been evaluated using cross-dataset setting on both `ASVspoof-2019` and `Fake or Real` datasets are selected for performance comparison in section 6.

Despite these advancements, there is limited research involving cross-dataset evaluations for fake audio detection. Many SOTA models, while effective on the datasets they were trained on, struggle with generalizing to unseen TTS or VC models (Chen et al., 2020). This generalization problem is particularly concerning as synthetic audio generation technologies continue to evolve, producing increasingly realistic audio that can evade detection. Therefore, enhancing the generalization ability of detection models is critical for improving the security of voice-based systems. Given these challenges, this work investigates the potential of SNNs to address the generalization issue in fake audio detection, particularly in scenarios involving unseen TTS and VC models.

**Partial Fake Audio Detection** Partial fake audio consists of a mixture of fake and real utterances, making it particularly difficult for deep learning models to detect. The existing models in literature, typically trained on datasets containing entirely fake or entirely real samples, struggle to identify the manipulated portions when genuine audio is present (Rahman et al., 2022). This limitation arises because most current models are designed for binary classification and they lack the granularity to detect individual fake/real segments within a single audio file. Time-variant models, such as those based on DNNs with variable input and output lengths, have been proposed to address this challenge. For instance, Zhang & Sim (2022) implements a three-stage approach to localize partial fake segments within an audio sample. While this approach shows promise, its multi-stage nature introduces latency, making it less suitable for real-time applications where fast processing is rather important. Furthermore, there is a significant lack of open-source datasets containing diverse range of attacks designed by utilizing partial fake audio, which limits the development and evaluation of more advanced models capable of detecting specific manipulated segments. Currently, the `PartialSpoof` dataset (Zhang et al., 2021), which was created using the `ASVspoof-2019` dataset, is the only publicly available dataset containing partially fake audio. However, `PartialSpoof` dataset also labels partially fake audio as fully fake audio instead of labeling individual segment of audio into fake or real.

**Spiking Neural Networks** Recently, SNNs have gained attention as a biologically inspired alternative to traditional ANNs due to their temporal dynamics and energy efficiency (Yamazaki et al., 2022). Due to recurrent nature of spiking neurons, SNNs are well-suited for handling temporal data, and have been successfully applied to tasks such as sound localization and classification (Baek & Lee, 2024). Convolution-based SNNs, in particular, have demonstrated strong performance in image processing by combining the strengths of both ANNs and SNNs (Mozafari et al., 2019; Zhou et al., 2020; Kirkland et al., 2020). While SNNs have proven effective in sound-related tasks, their application to fake or partially fake audio detection remains largely unexplored. This work aims to bridge that gap by investigating their potential in this area.

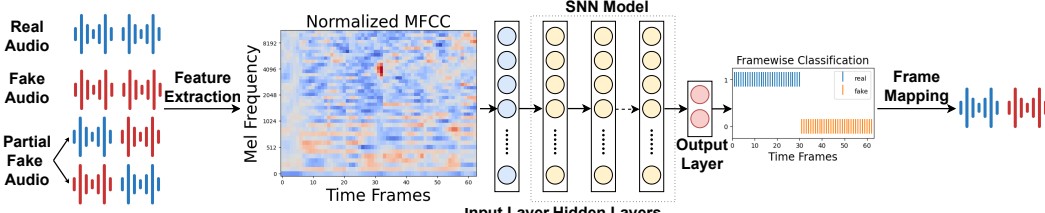

Figure 1: Overview of the proposed SNN based approach for fake and partial fake audio detection.

## 3 PRELIMINARIES

SNNs can be represented using various models, such as the *Leaky Integrate-and-Fire (LIF)* model, Hodgkin-Huxley model, and Spike Response Model, each capturing distinct aspects of neuronal dynamics and behavior. Given its proven effectiveness for power-efficient deep learning (Rozenberg et al., 2019), our work employs the LIF model for implementing SNNs.

**Leaky Integrate and Fire Neuron** The LIF neuron is a simplified model of a biological neuron, widely used in computational neuroscience to simulate the electrical activity of neurons in a network (Dayan & Abbott, 2001). The LIF neuron has a membrane potential $U(t)$ which increases with input $I(t)$ (synaptic current or stimulus) and decay with membrane potential decay rate $\beta$. The neuron "fires" or generates a spike when the membrane potential reaches a certain threshold and resets its membrane potential according to reset mechanism. Popular reset mechanisms include subtracting with threshold potential and setting membrane potential to zero. The membrane potential of a neuron can be described by the following equation:

$$U(t + 1) = \beta \times U(t) + I(t + 1) - R(\beta \times U(t) + I(t + 1)) \tag{1}$$

where $R$ is the reset mechanism for reset to zero. $R$ is set to 1 when the neuron fires, and 0 otherwise.

**Surrogate Gradient Descent** Training SNNs through supervised learning is challenging due to the discrete nature of spikes. During the forward pass, spikes are represented using a shifted Heaviside step function. To calculate gradients (partial derivative of the loss with respect to parameters) during the backward pass, spikes are approximated using a smooth surrogate functions such as *Fast Sigmoid (FS)* (Zenke & Ganguli, 2018) and *Arctangent* (Fang et al., 2021).

**Loss Functions** To train SNN models for classification task, two commonly used loss functions for backpropagation are *Cross Entropy Spike Count (CE-count)* and *Cross Entropy Rate (CE-rate)*. The CE-count loss function calculates the total spike count over time for the output neurons of each class. The predicted spike counts are compared to the target spike counts, which are derived by multiplying the ground truth labels by the number of time steps. These values are then passed through the CE function to compute the loss. This approach encourages consistent spiking of the correct class throughout the time steps while minimizing spikes from incorrect classes. On the other hand, the CE-rate loss function processes spike outputs sequentially at each time step. At each time step, the spike outputs and the corresponding ground truth values are passed through the CE function, with the resulting losses accumulated over time. Similar to CE-count, CE-rate promotes consistent spiking of the correct class and suppresses incorrect spikes, but it does so by considering spike activity at each individual time step rather than across the entire time sequence.

## 4 METHODOLOGY

Figure 1 provides a high-level overview of the proposed approach. The key components of the approach include the datasets, feature extraction, and classification using the proposed model.

### 4.1 FEATURE EXTRACTION

The raw audio samples in the datasets that we use in our experiments (detailed in section 4.4) are $2\,\mathrm{seconds}$ in length and sampled at a rate of $16\,\mathrm{kHz}$, resulting in 32,000 floating point values per sample. Directly passing these floating points as an input can significantly increase the number of

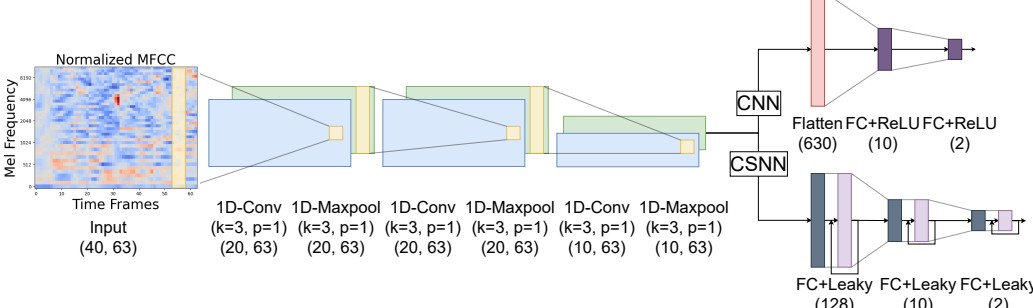

Figure 2: CNN and CSNN model architecture.

parameters in the input layer of the neural network models, leading to computational inefficiencies. To mitigate this, we pass frequency-temporal features - Mel-Frequency Cepstral Coefficients (MFCCs) as input to the models. MFCCs are computed by transforming an audio signal into the power spectrum using Short Term Fourier Transform (STFT), applying a Mel filter bank, taking the logarithm of the filter energies, and then performing a discrete cosine transform (DCT) to extract the most relevant coefficients. In this study, for the STFT, a window length of 2048 samples and a hop length of 512 samples (25% overlap) are used. The input is zero-padded on both sides to ensure that each frame of the STFT output is centered with the corresponding position in the original signal. Consequently, the output contains 63 time frames (columns), each representing $128\,\text{milliseconds}$ of input audio. From the Mel-filtered spectrum, 40 MFCCs are extracted, resulting in 40 channels (rows) in the input features. Lastly, MFCCs are normalized using Lp-norm normalization, preventing those with larger scales from disproportionately influencing the learning process.

## 4.2 ARTIFICIAL NEURAL NETWORK MODELS

Due to the lack of cross-dataset evaluation studies on assessing the generalization ability of ANNs for fake audio detection problem, we implemented three representative ANN models, MLP, CNN, and TE. This aims to establish baseline performance by ANN models that utilize similar resources, including datasets, as those employed by SNN models. Model details are as follows.

**Multi Layer Perceptron** We implement a 5-layer Fully-Connected Feed-Forward Neural Network (FC-FFNN). The layers contain 2520 (input), 256, 128, 10, and 2 (output) neurons, respectively. A Rectified Linear Unit (ReLU) activation function is used at each hidden layer to introduce non-linearity allowing the model to learn complex patterns. The input to the network consists of flattened MFCCs of size $40 \times 65 = 2520$. The output of the final layer is used for binary classification (real or fake audio).

**Convolutional Neural Network** We then implement a CNN model as demonstrated in fig. 2. The convolution layer in the model uses 1D convolutional and 1D max-pooling, applied over the time domain, to extract deep features from the MFCCs of input audio. 40 MFCCs are treated as 40 input channels. The subsequent convolutional layers have 20, 20, and 10 channels, respectively. In both the convolutional and max-pooling layers, a stride of one and padding (zero-padding) of one is used to preserve the temporal dimensions of the data. The output from the final max-pool layer is flattened into a 630-dimensional vector and fed into a FC-FFNN for classification. The FC-FFNN consists of three layers with 630, 10, and 2 neurons, respectively, where the ReLU activation function is applied to the intermediate layers. The output of the final layer is used for binary classification (real or fake audio).

**Transformer Encoder** We also implement a TE model which uses the self attention mechanism proposed by Vaswani (2017) to encode input MFCCs. Encoding layer consist of 6 encoding blocks, each with 8 attention heads and a feed-forward network with dimension 1024. The encoded input maintains the same shape as the input MFCCs ($40 \times 63$). Flattened encoded input is than passed through FC-FFNN for classification. FC-FFNN consist of 4 layers with 2520 (flattened), 1024, 10 and 2 (output) neurons with ReLU activation function in between the hidden layers. Similar to previous ANN models, the final output layer is used for binary classification.

### 4.3 Spiking Neural Network Models

**SNN** Compared to traditional ANNs, SNNs differ in the way they process and transmit information. Instead of using continuous values to represent activation, SNNs communicate via discrete spikes, where each spike represents a binary value of 1 (spike) or 0 (no spike). In this work, for the purpose of detecting fake and partial fake audio, we propose a SNN model consisting of an input layer with 40 neurons, followed by four spiking layers containing 256, 126, 10, and 2 (output) neurons, respectively. Each spiking layer comprises a Fully Connected (FC) layer from an ANN model and a corresponding Leaky layer. The leaky layer consists of LIF neurons, which are connected one-to-one with the neurons in the preceding FC layer, similar to the ReLU activation function in ANN models. The leaky layer serves as an activation mechanism, outputting either a spike (1) or no spike (0). We set the decay parameter ($\beta$) for the LIF neurons to 0.9, and the spike threshold is learned during training. The input is processed sequentially, with the 40 MFCCs from one time frame passed into the network at a time. This sequential input allows the SNN model to capture temporal dependencies, making it independent of input length. Additionally, the membrane potential of the LIF neurons is preserved across time frames, introducing a recurrent component that enables the model to retain information over multiple time steps. These temporal and recurrent properties allow the SNN to effectively model the dynamic nature of audio signals. In SNN, input audio is classified based on spike count of two output neurons for binary classification. For the partial fake audio detection problem, each time frame is classified based on output of two output neurons at each time step.

**Convolutional SNN** While models such as the TE are computationally and power-intensive, simpler models such as MLP may lack the complexity needed to effectively capture subtle patterns in large audio datasets (Müller et al., 2022). CNNs strike a balance by efficiently extracting complex features through convolutional layers while utilizing smaller fully connected layers for classification. On the other hand, while SNNs are generally energy efficient, they may also lack complexity to fit diverse datasets. To this end, we propose a novel Convolutional Spiking Neural Network (CSNN) based approach that combines the feature extraction power of CNNs with the temporal processing capabilities of SNNs for the task of fake and partial fake audio detection. As shown in fig. 2, CSNN retains the CNN architecture up to the final maxpool layer, where deep features are extracted from the MFCCs. These deep features are then passed through three spiking layers containing 128, 10, and 2 neurons, respectively. Similar to the earlier SNN model, we set the decay parameter ($\beta$) to 0.9, and the spike threshold is learned during training. Fake and partial fake audios are classified using mechanism similar to the SNN model.

### 4.4 Datasets

To evaluate the proposed fake audio detection models, we use two publicly available datasets, `ASVspoof-2019` (Wang et al., 2020b) and `Fake or Real` (Reimao & Tzerpos, 2019). The primary motivation for selecting multiple large-scale datasets is to evaluate and compare the generalization capabilities of the ANNs and the proposed SNN models.

**ASVspoof-2019** The first ASVspoof dataset was released as a part of Automatic Speaker Verification (ASV) challenge in 2015 (Wu et al., 2014). Updated versions of the dataset are released every two years, with the latest being `ASVspoof-2021` (Yamagishi et al., 2021). In this study, we utilize the `ASVspoof-2019` dataset, as it specifically focuses on spoofing countermeasures. The `ASVspoof-2019` dataset is divided into two subsets: Logical Access (LA) and Physical Access (PA). The LA subset addresses spoofing attacks where the attacker can access the target device remotely, while the PA subset focuses on attacks where the attacker has physical access to the device. For this study, we focus exclusively on the LA subset (referred to as `ASVspoof` for the remainder of the paper), where spoofed samples are generated using 4 TTS and 2 VC models for the training and validation sets followed by 7 TTS and 6 VC models for testing set (Wang et al., 2020b). The audio samples in this dataset are sampled at $16\,\text{kHz}$.

Figure 6 (in appendix A) shows the distribution of sample lengths for both fake and real audio samples in the dataset, indicating that the average length of both fake and real samples are around $3\,\text{seconds}$. To ensure consistency across both the datasets used in this study, all audio samples in the dataset are standardized to a length of $2\,\text{seconds}$. Samples longer than $2\,\text{seconds}$ are trimmed, while shorter samples are padded with zeros (silence) to match the required length. As it can be seen in

table 3 (in appendix A) and fig. 6, there is a severe class imbalance between real and fake samples in the `ASVspoof` dataset. To mitigate this imbalance, we reduce the fake samples in the training set to 2,580 (by randomly selecting 430 samples from each of the 4 TTS and 2 VC spoofing models used in the training set).

**Fake or Real** This dataset (detailed in appendix A.1) was designed for the evaluation of spoofing countermeasures in ASV systems (Reimao & Tzerpos, 2019). For our experiments, we utilize the `FoR-2seconds` (referred to as `FoR` for the remainder of the paper) variation of the dataset as it provides samples of uniform length of 2 seconds, ensuring consistency. As it can be seen in table 3 (in appendix A) the distribution of samples in the training, validation, and testing sets of the `FoR` dataset are perfectly balanced. The testing set contains previously unseen fake samples (generated using Google Cloud TTS with Wavenet), along with previously unseen real samples. To further assess whether the proposed models can adapt TTS audio generated using new algorithms, inspired by Reimao & Tzerpos (2021) we created the `FoR-mix` dataset. The `FoR-mix` dataset is constructed by removing 200 randomly selected samples from the test set and adding them into the training set of the `FoR` dataset.

**Partial Fake Audio Dataset** We then create a partial fake-audio dataset (`PFA` dataset) to assess the performance of the baseline ANN models and our proposed SNN models in the presence of partial fake audio. Additionally, we utilize this newly created dataset to train SNN models to classify audio inputs by segmenting them into shorter temporal frames, rather than classifying the entire audio sample as a whole. The ground truth of each audio sample is created by aggregating the ground truth of each time frame. This approach also eliminates the constraint of classifying an entire audio sample based on a fixed initial length. For example, an adversary could attempt to bypass spoofing detection by appending fake audio after an initial segment of real audio. By segmenting the audio into shorter temporal frames and classifying them all, a model can analyze the entire audio sample, improving its ability to detect such spoofing attempts and making it more robust against partial fake audio based evasion strategies. The dataset consists of four types of audio samples: (1) two-second fake, (2) two-second real, (3) one second fake followed by one second real, and (4) one second real followed by one second fake. Samples are generated using fake and real samples from the `FoR` dataset.

The training, validation, and test sets are constructed using samples explicitly from the training, validation, and test sets of the `FoR` dataset. Table 4 (in appendix A.3) shows the balanced class distribution in the newly constructed `PFA` dataset. To achieve the objective of improving generalization to TTS audio generated by new algorithms, we applied a similar strategy (to `FoR-mix` dataset creation) by removing 800 randomly selected samples from the test set and adding them to the training set.

## 5 EXPERIMENTS

We design three experiments to assess the proposed SNN and CSNN model performance in fake and partial fake audio detection. All experiments were run on a Nvidia L40S GPU and implementation details are provided in appendix B. The experiments are evaluated based on two key metrics: Equal Error Rate (EER) and accuracy (as detailed in appendix C).

### 5.1 EXPERIMENT 1: HYPERPARAMETER TUNING

This experiment aimed to identify the optimal hyperparameters for SNN and CSNN models in fake audio detection. The hyperparameters tested included two loss functions—CE-rate and CE-count—and two surrogate gradients—Fast Sigmoid (FS) and Arctangent. Both SNN and CSNN models were trained with all four combinations of these loss functions and surrogate gradients, using the `ASVspoof` and `FoR` datasets independently. For each dataset, the optimal hyperparameters were selected based on model performance on the validation set. These selected models were then utilized in subsequent experiments.

## 5.2 EXPERIMENT 2: FAKE AUDIO DETECTION

Spoofed or fake data generated by newer and more advanced TTS or VC models can easily evade detection in older spoofing detection systems (Müller et al., 2022). Consequently, it is essential to assess spoofing detection model's ability to identify and adapt to spoofed audio generated by spoofing algorithms that were not encountered during training. To address this challenge, this experiment assesses the generalization capability of SNN and CSNN models for fake audio detection by testing their performance on datasets different from those used during training. To this end, we train and test the baseline ANN models and the proposed SNN models on the `ASVspoof` and `FoR` datasets in a cross-dataset setting. Spoofed samples in the `ASVspoof` test set are created using 11 unknown spoofing models (TTS and VC) and 2 known spoofing models (TTS and VC). Whereas, in the `FoR` test set, fake samples are created using previously unseen more commercialized sourced services such as Google Cloud TTS with WaveNet (Reimao & Tzerpos, 2019). Furthermore, the real utterances in both test sets are distinct from the real utterances in both train sets. This type of diversity in test sets further challenges proposed models' generalization ability. Additionally, to evaluate proposed models' ability to learn spoofed samples generated using previously unseen spoofing algorithms, proposed models are trained and evaluated using `FoR-mix` dataset (see section 4.4).

To evaluate SNN model performance on the `FoR` dataset, we use the implemented ANN models and the Stacked Temporal Convolution Network (Stacked TCN) model proposed by Firc et al. (2024) (which provides a similar cross-dataset evaluation). Additionally, we compare against three models presented by Reimao & Tzerpos (2021), which uses the `FoR` dataset. These models include Random Forest (RF) using MFCC features, RF using CQT features, and VGG-19 using STFT features. The first two models represent the SOTA, non-deep-learning approaches, while the VGG-19 model provides a benchmark for deep learning. For the `ASVspoof` dataset, we compare our results with the performance of Baseline 1 and Baseline 2 models by Wang et al. (2020b), as well as the Stacked TCN model by Firc et al. (2024).

## 5.3 EXPERIMENT 3: PARTIAL FAKE AUDIO DETECTION

This experiment proposes a novel approach that utilizes temporal nature of SNN models to detect partial fake audio by classifying audio at the individual time frame level. We use our newly created `PFA` dataset (see section 4.4) to train the proposed SNN and CSNN models. We use the CE-rate as the loss function due its ability to compare the predicted output of individual time frames with their respective ground truths. Based on the hyperparameter tuning (see section 6.1), we choose Arctangent as the surrogate gradient. The accuracy is then determined by the the number of correctly predicted time frames over the total number of time frames. To assess how training on the `PFA` dataset could enhance the models' ability to detect partial fake audio, we compare the performance of these models against SNN and CSNN models with similarly configured hyperparameters trained on the `FoR-mix` dataset. To further demonstrate the vulnerability of ANN and SNN models to partial fake audio, we evaluate the baseline ANN models and SNN models, trained on the `FoR-mix` dataset for fake audio detection, by testing them on `PFA` dataset. In this experiment, only fully real audio samples are categorized as real, while both fully fake and partially fake samples are classified as fake.

## 6 RESULTS

### 6.1 EXPERIMENT 1: HYPERPARAMETER TUNING

Figure 3a and fig. 3b illustrate the performance of the SNN and CSNN models with different loss functions and surrogate gradients on the `FoR` and `ASVspoof` datasets, respectively. Both models demonstrated superior performance with the CE-count loss function compared to the CE-rate loss across both datasets. While the models performed similarly with both surrogate gradients, the SNN achieved the highest validation accuracy of 99.47% with the FS surrogate gradient on the `FoR` dataset, while the CSNN achieved 99.04% with the Arctangent surrogate gradient. On the `ASVspoof` dataset, performance was measured using the EER metric due to the class imbalance in the validation set (table 3). The SNN achieved the lowest validation EER of 6.28% with the Arctangent surrogate gradient, and the CSNN achieved 5.73% with the FS surrogate gradient. Across all experiments, the CE-count loss consistently outperformed or matched the CE-rate loss. The CE-

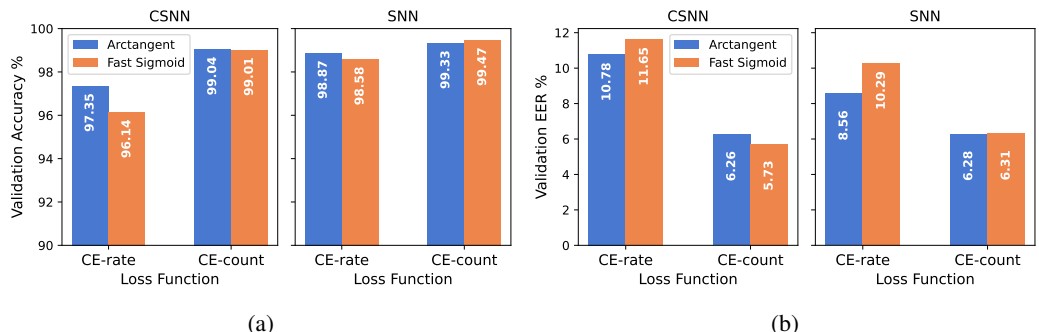

(a)                          (b)

Figure 3: SNN and CSNN model hyperparameter tuning for (a) `FoR` and (b) `Asvspoof` datasets.

count loss applies the loss only once after accumulating spikes, making it particularly effective for classifying entire audio sample. In contrast, the CE-rate loss, which applies the loss at every time step, is better suited for classifying individual time frames.

**Summary:** SNN and CSNN models demonstrated superior performance on both the `ASVspoof` and `FoR` datasets using the CE-count loss. Both Arctangent and FS surrogate gradients showed comparable results across both models and datasets.

## 6.2 EXPERIMENT 2: FAKE AUDIO DETECTION

Table 2: Cross-dataset testing after training using `FoR` and `ASVspoof`. Missing values indicate that either the respective work did not present those metrics or did not perform those specific analysis.

| Model | Parameters | Trained on FoR | | | | Trained on ASVspoof | | | |
|---|---|---|---|---|---|---|---|---|---|
| | | FoR | | ASVspoof | | ASVspoof | | FoR | |
| | | Accuracy | EER | Accuracy | EER | Accuracy | EER | Accuracy | EER |
| MLP | 679,584 | 82.35 | 17.61 | 18.63 | 64.52 | 84.36 | 13.57 | 50.18 | 46.32 |
| CNN | 8,472 | 71.97 | 29.50 | 35.84 | 34.11 | 87.52 | 11.74 | 53.86 | 50.55 |
| TE | 3,130,000 | 64.80 | 20.04 | 27.30 | 48.16 | 85.21 | 11.38 | 50.00 | 40.26 |
| SNN | 44,708 | 54.96 | 29.23 | 12.93 | 55.35 | 83.54 | 12.98 | 50.00 | 69.39 |
| CSNN | 6,063 | 71.60 | 22.70 | 24.78 | 31.18 | 84.00 | 12.25 | 50.00 | 54.59 |
| RF-MFCC (Reimao & Tzerpos, 2021) | - | 56.98 | - | - | - | - | - | - | - |
| RF-CQT (Reimao & Tzerpos, 2021) | - | 86.94 | - | - | - | - | - | - | - |
| VGG-19-STFT (Reimao & Tzerpos, 2021) | - | 52.02 | - | - | - | - | - | - | - |
| Stacked TCN (Firc et al., 2024) | - | - | 6.99 | - | 47.73 | - | 23.37 | - | 46.04 |
| `ASVspoof`-Baseline1 (Wang et al., 2020b) | - | - | - | - | - | - | 9.57 | - | - |
| `ASVspoof`-Baseline2 (Wang et al., 2020b) | - | - | - | - | - | - | 8.09 | - | - |

As shown in table 2, models trained on the `FoR` dataset and tested on the `ASVspoof` dataset demonstrate a decline in accuracy as model complexity increases. This suggests that more complex ANN models are prone to overfitting, which leads to reduced generalization. In contrast, the SNN and CSNN models achieved comparable performance to the ANN models but with fewer parameters. The CSNN model, in particular, achieved the lowest EER (31.18%) during cross-dataset evaluation on the `ASVspoof` test set, outperforming the Stacked TCN (Firc et al., 2024) model by 16.55%.

When trained on the `ASVspoof` dataset (table 2), all models performed similarly on both the `ASVspoof` and `FoR` test sets. However, the SNN and CSNN models showed improvements of 10.39% and 11.12%, respectively, over the stacked TCN model. The higher EERs observed on the `FoR` test set highlight the difficulty in detecting fake audio generated using more advanced and previously unseen TTS models. As illustrated in fig. 4, models trained on the `FoR-mix` dataset showed significant improvements in test accuracy compared to those trained on the `FoR` dataset. This suggests that the generalization ability of both ANN and the proposed SNN models is heavily influenced by the diversity and quality of the training data.

**Summary:** The SNN and CSNN models performed comparably to baseline ANN models while using fewer parameters. The CSNN model demonstrated better cross-dataset generalization after training on the `FoR` dataset, outperforming both ANN and SOTA models. Training on the `FoR-mix` dataset further enhanced the generalization capabilities of both SNN and ANN models, highlighting the importance of diverse training data.

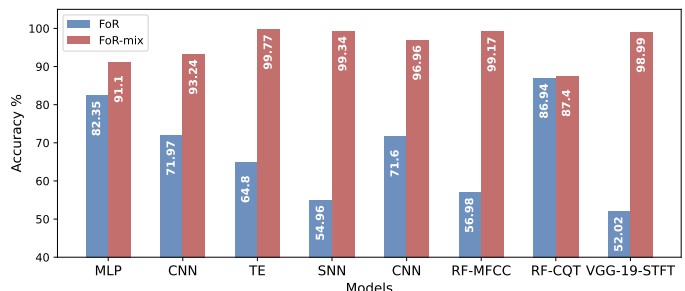

Figure 4: Model performances on `FoR` and `FoR-mix`.

### 6.3 EXPERIMENT 3: PARTIAL FAKE AUDIO DETECTION

As shown in fig. 5 (left), all models trained on the `FoR-mix` dataset struggled to detect partial fake audio in the `PFA` dataset. Figure 5 (right) highlights the performance improvements of SNN and CSNN models before and after training on the `PFA` dataset. The proposed SNN model achieved an accuracy of 83.98% on `PFA` dataset, reflecting a 10.59% improvement compared to its performance after training only on the `FoR-mix` dataset. Similarly, the CSNN model reached 85.59% accuracy, representing a 15.94% improvement over its performance on `FoR-mix` and a 1.61% improvement over the SNN model trained on the `PFA` dataset. These results suggest that while SNN models trained to classify individual time frames using the CE rate loss and `FoR-mix` dataset perform better than traditional ANN models, their performance can be further enhanced by training them on a dataset that contains partial fake audio. The higher accuracy of the CSNN model indicates that SNN models are strong candidates for partial fake audio detection when classifying short temporal frames.

**Summary:** Both ANN and SNN models experienced a drop in accuracy when tested on the `PFA` dataset after training on the `FoR-mix` dataset. However, after training on the `PFA` dataset, the CSNN and SNN models showed significant performance improvements.

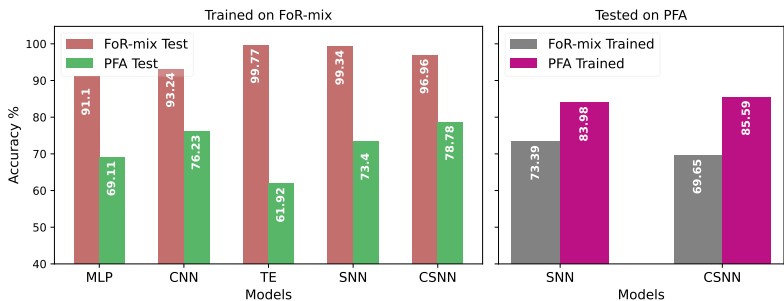

Figure 5: Model performance on `PFA` dataset when trained on `FoR-mix` dataset (left), and trained on `PFA` dataset (right).

## 7 CONCLUSION

In this paper, we explored the use of SNNs for the detection of fully and partially synthesized fake audio, addressing a gap in existing research. Our experiments demonstrated that SNN models, particularly the CSNN, achieved performance comparable to ANNs and other SOTA models with considerably lower number of parameters. Despite this efficiency, the SNN models did not demonstrate substantial improvements in generalization in the presence of fake audio generated by previously unseen algorithms. However, the temporal dynamics inherent to SNNs enabled a novel approach for detecting partial fake audio at the frame level, offering a promising direction for future advancements. This work lays the foundation for future research into enhancing the robustness and generalization of SNN models, especially in security-critical audio manipulation detection.

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

## APPENDIX

## A  FURTHER DETAILS ON DATASETS

### A.1  FOR

The `FoR` dataset is available in four variations: `FoR-original`, `FoR-norm`, `FoR-2seconds`, and `FoR-rerecorded`. `FoR-original` contains the original, unprocessed samples. `FoR-norm` consists of samples that have been converted to WAV format, normalized to 0 dBFS, downsampled to a 16 kHz sample rate, and converted to mono. Additionally, silences at the beginning and end of the utterances have been removed. `FoR-2seconds` includes the `FoR-norm` samples truncated to 2 seconds in length, while `FoR-rerecorded` comprises re-recorded utterances to simulate real-world attacks.

### A.2  ASVSPOOF

Figure 6 shows the distribution of sample lengths for fake and real audio samples in the `ASVspoof` dataset, which shows that the average length of both fake and real samples are around 3 seconds. Figure 6 also shows the severe class imbalance that exists in real vs. fake samples in the `ASVspoof` dataset (also see table 3).

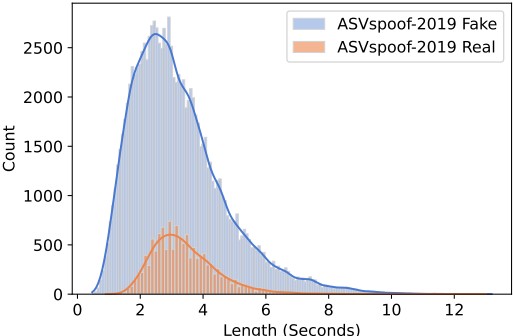

Figure 6: Class-wise audio sample length distribution in ASVspoof-2019 dataset.

Table 3: Class distribution in ASVspoof and FoR datasets.

| Datasets | Train | | Validation | | Test | |
|---|---|---|---|---|---|---|
| | Real | Fake | Real | Fake | Real | Fake |
| ASVspoof (Wang et al., 2020b) | 2,580 | 22,800 | 2,548 | 22,296 | 7,355 | 63,882 |
| Fake or Real (Reimao & Tzerpos, 2019) | 6,978 | 6,978 | 1,413 | 1,413 | 544 | 544 |

### A.3  PFAD

Table 4 shows the balanced class distribution in our newly constructed `PFA` dataset which aids in ensuring that the evaluated models does not become biased toward any particular class, improving their ability to generalize.

Table 4: Class distribution in Partial Fake Audio dataset `PFAD`.

| Partial Fake | Train | Validation | Test |
|---|---|---|---|
| Fake | 7,178 | 1,413 | 344 |
| Fake+Real | 7,178 | 1,413 | 344 |
| Real+Fake | 7,178 | 1,413 | 344 |
| Real | 7,178 | 1,413 | 344 |
| **Total** | 28,712 | 5,652 | 1,376 |

## B  Implementation

All our experiments were implemented in Python 3.11 on a server running Ubuntu 22.04 with 350 GB RAM and a Nvidia L40S (48 GB VRAM), 48 CPU cores, and 400 GB swap memory) GPU. ANN models were built using the PyTorch library (Paszke et al., 2019), while SNN models were developed using PyTorch alongside snnTorch (Eshraghian et al., 2023). All our proposed models and baseline models were trained for 200 epochs, and the epoch with the minimum validation loss was selected for evaluation. The ANN models were trained with CE loss function, the Adam optimizer with learning rate of 0.0001 and an L2-penalty of 0.000005 to reduce over-fitting. The SNN models were trained with Adam optimizer and learning rate of 0.0005.

## C  Metrics

We use the following metrics to evaluate the performance of our proposed models.

1. **Accuracy**: The model's accuracy is measured as the proportion of correctly classified instances out of the total instances provided to the model.

2. **Equal Error Rate (EER)**: The point where the *False Positive Rate (FPR)* (proportion of real audio incorrectly classified as fake) equals the *False Negative Rate (FNR)* (proportion of fake audio incorrectly classified as real). A lower EER is an indicator of a more accurate and balanced model.

In the context of fake audio detection, accuracy can be misleading when dealing with imbalanced datasets. Therefore, we use EER as a more reliable metric, as it balances the trade-off between FPR and FNR. EER is particularly useful in situations where both types of errors (failing to detect fake or real audio) are critical.

## D  Additional Results

### D.1  Experiment 1: Hyperparameter Tuning

Table 5 shows the validation accuracies and EERs obtained during the hyperparameter tuning on the `ASVspoof` dataset. While accuracy provides a general overview of model performance, the EER is a more reliable metric in this case due to the class imbalance in the `ASVspoof` dataset, as it equally considers both false positives and false negatives, offering a clearer picture of detection performance. This is evident in table 5, where certain hyperparameters with similar accuracies have notably different EERs.

Table 5: SNN and CSNN validation results for `ASVspoof`.

| | SNN | | | | CSNN | | | |
| | Fast Sigmoid | | Arctangent | | Fast Sigmoid | | Arctangent | |
| | Accuracy % | EER % | Accuracy % | EER % | Accuracy % | EER % | Accuracy % | EER % |
|---|---|---|---|---|---|---|---|---|
| **CE-rate** | 94.86 | 10.29 | 96.15 | 8.56 | 94.83 | 11.65 | 95.31 | 10.78 |
| **CE-count** | 96.91 | 6.31 | 96.99 | **6.28** | 97.11 | **5.73** | 97.17 | 6.26 |

### D.2  Experiment 2: Fake Audio Detection

Figure 7 illustrates the trade-off between the number of parameters and accuracy for various models. Notably, both SNN and CSNN achieved comparable performances with relatively low number of parameters, making them more efficient compared to models such as Transformers, which require significantly more parameters for a comparable level of accuracy.

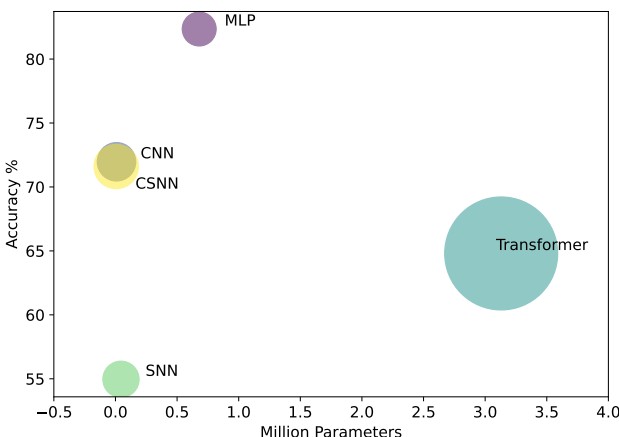

Figure 7: Number of parameters vs. the accuracy vs. model depth (indicated by the radius).

## E  FURTHER DISCUSSION

Our study highlights key areas for improvement in the development of fake audio detection models, particularly in terms of generalizability. One major challenge revealed in our results is that the models, including both SNNs and ANNs, often fail to generalize well when tested on data from unseen voice synthesizing algorithms. This suggests that the models are overfitting to the specific patterns present in the training data and struggle to adapt to novel manipulations introduced by new algorithms. However, when the training set includes samples from these unseen algorithms, model performance improves considerably. This indicates that the limited generalization is not necessarily a failure of the model architecture itself, but rather a limitation of the diversity of the training data.

In adversarial settings, one of the key challenges is the unpredictability of the specific algorithm used to generate synthetic audio, making it difficult for detection models to generalize effectively. Attackers may use novel or customized voice synthesizing techniques that the model has never encountered, resulting in significant detection blind spots. This is particularly problematic because the rapid pace of advancements in TTS and VC technologies means new, highly realistic algorithms are constantly emerging, further complicating the task for existing models. To address this challenge, it is essential to maintain a continuously evolving, comprehensive dataset that captures a wide array of known voice synthesizing algorithms. Another key area for further development is the creation of more sophisticated partial fake audio datasets. There is a need for a more advanced dataset that captures a wider range of manipulations, including more complex synthetic audio generation techniques. Moreover, making such a dataset publicly available, with frame-level annotations for real and fake segments, would allow other researchers to benchmark their models and drive progress in this domain.

A potential direction for future research is to compare the power efficiency of SNNs and traditional ANNs by implementing SNNs on neuromorphic hardware platforms, such as Intel Loihi (Davies et al., 2018). Such a comparison could provide valuable insights into the practical advantages of SNNs over conventional ANNs in fake audito detection, especially in large-scale deployment scenarios.