# OpenReview forum: "SAFE: Spiking Neural Network-based Audio Fidelity Evaluation"
_ICLR.cc/2025/Conference — ICLR 2025 Conference Withdrawn Submission_

### Official Review · Reviewer_e3uV · 2024-11-03

**Soundness:** 3
**Presentation:** 3
**Contribution:** 2
**Rating:** 5
**Confidence:** 4

**Summary:**

The paper proposes a fake audio detection method based on SNN, aimed at detecting fully and partially fake audio. By applying SNN to this task, the paper demonstrates the potential advantages of this approach in terms of energy efficiency and model parameter reduction. The authors develope two types of SNN models (a fully connected SNN and a convolutional SNN) and evaluate their generalization performance across various datasets, such as ASVspoof and Fake or Real. Additionally, a partially fake audio dataset is created to test the models' fine-grained detection capabilities.

**Strengths:**

1. This study is the first to explore the use of SNN for audio deepfake detection, introducing a novel neural network architecture choice to the field.

2. Two SNN-based models are proposed in this research, demonstrating superior performance in parameter efficiency and energy consumption compared to traditional ANN, making them more suitable for resource-constrained applications.

3. Although the number of datasets tested is limited, the study provides a comprehensive evaluation of the approach, including hyperparameter tuning, cross-dataset testing, and detection of partially fake audio, thoroughly assessing the model's performance.

**Weaknesses:**

1. The application of SNN for audio deepfake detection brings a degree of novelty; however, simply employing SNNs for this task may limit the paper's depth of contribution. A more detailed explanation is needed to justify the choice of SNNs for audio deepfake detection, particularly considering the success of ANN-based models in this area. It is recommended to design and optimize the SNN model in alignment with the specific characteristics of audio deepfake detection to demonstrate an innovative approach beyond the straightforward application of SNN.

2. The dataset tested in this paper is limited, which can not reflect the effectiveness and generalization of the method proposed in this paper.

3. The comparative analysis is insufficient, especially in terms of comparisons with the latest deepfake detection methods.

4. The descriptions of SNN architecture and implementation details are relatively brief. Including more detailed explanations would enhance the method's interpretability and reproducibility.

**Questions:**

1.It is hoped that the architecture and design of SNN model can be further improved in combination with the characteristics of audio depfake detection, rather than simple application.

2.To enhance the clarity of the paper's contributions, the authors could address the following aspects:

a.Conduct additional experimental analysis and comparisons to demonstrate the necessity of using SNN models for deepfake tasks.

b.Compared with the direct application of SNN, the model in this paper can reduce the amount of calculation and required parameters, which is the advantage of SNN model, and can not explain the contribution of the method in this paper. It needs to be compared with the direct application to illustrate how the improvement of the architecture in this work can further improve the performance and increase the generalization ability.

3.Provide more implementation details, such as pseudocode descriptions, to improve reproducibility.

---

> ### Author Response · Authors · 2024-11-19
>
> Thank you for your insightful review of the submitted paper. Below, we address your questions and outline the updates that will be incorporated into the revised version.
>
> 1. **Feedback**: The application of SNN for audio deepfake detection brings a degree of novelty; however, simply employing SNNs for this task may limit the paper's depth of contribution. A more detailed explanation is needed to justify the choice of SNNs for audio deepfake detection, particularly considering the success of ANN-based models in this area. It is recommended to design and optimize the SNN model in alignment with the specific characteristics of audio deepfake detection to demonstrate an innovative approach beyond the straightforward application of SNN.
>
>     **Response**: As mentioned in the introduction section “SNNs are inherently designed to process temporal data due to their ability to capture the timing and sequence of events, which makes them particularly suitable for tasks involving audio, which has rich temporal dynamics (Baek & Lee, 2024). By leveraging these capabilities, our work explores the feasibility of using SNNs for fake audio detection and comprehensively evaluate their performance across various tasks, including cross-dataset generalizability evaluation and the detection of partial fake audio.” However, we will provide a more detailed justification as to why we explored the use of SNNs for the task of fake audio detection in the revised submission.
>
> 2. **Feedback**: The dataset tested in this paper is limited, which can not reflect the effectiveness and generalization of the method proposed in this paper.
>
>     Response: We acknowledge that the current datasets may not fully capture the breadth needed to evaluate the generalization of the proposed method. To address your concern and further validate our models in real-world scenarios, we will include the In-the-Wild (Müller et al. 2022) dataset in the cross-dataset evaluation experiment in the revised submission. This dataset provides diverse and uncontrolled audio samples, which will better reflect the generalization capabilities and practical effectiveness of our approach.
>
> 3. **Feedback**: The comparative analysis is insufficient, especially in terms of comparisons with the latest deepfake detection methods.
>
>     **Response**: To address this, we will expand our comparative analysis by including advanced ANN models such as RawNet2 (Tak et al. 2021) and AASIST (Jung et al.2022) in Experiment 2. These models will be evaluated using cross-dataset testing to better assess their generalization performance alongside our proposed SNN and CSNN models.
>
> 4. **Feedback**: The descriptions of SNN architecture and implementation details are relatively brief. Including more detailed explanations would enhance the method's interpretability and reproducibility.
> Question: Provide more implementation details, such as pseudocode descriptions, to improve reproducibility.
>
>     **Response**: We agree that additional details would enhance the method's interpretability and reproducibility. While Section 4.3 (Spiking Neural Networks) provides an overview of the proposed models, we will expand on this by adding detailed descriptions of the architecture and the roles of key components, including Leaky, Convolution, MaxPool, and Fully-Connected layers.
>
> **References**:
>
> Baek, Suwhan, and Jaewon Lee. "Snn and sound: a comprehensive review of spiking neural networks in sound." Biomedical Engineering Letters 14.5 (2024): 981-991.
>
> Müller, N. M., Czempin, P., Dieckmann, F., Froghyar, A., & Böttinger, K. (2022). Does audio deepfake detection generalize? Interspeech 2022, Incheon, Korea, 18-22 September 2022.
>
> Tak, Hemlata, et al. "End-to-end anti-spoofing with rawnet2." ICASSP 2021-2021 IEEE International Conference on Acoustics, Speech and Signal Processing (ICASSP). IEEE, 2021.
>
> Jung, Jee-weon, et al. "Aasist: Audio anti-spoofing using integrated spectro-temporal graph attention networks." ICASSP 2022-2022 IEEE international conference on acoustics, speech and signal processing (ICASSP). IEEE, 2022.

---

### Official Review · Reviewer_e8Wk · 2024-11-03

**Soundness:** 2
**Presentation:** 3
**Contribution:** 2
**Rating:** 3
**Confidence:** 5

**Summary:**

This paper presents a novel approach to fake audio detection using Spiking Neural Networks (SNNs), specifically focusing on the challenges of detecting both fully and partially synthetic audio. The study's strengths include the development of an energy-efficient SNN model with comparable accuracy to state-of-the-art Artificial Neural Networks (ANNs) and its unique application to partial fake audio detection at the frame level, which is underexplored in existing research. Another strength is the detailed experimental analysis, which includes hyperparameter tuning and cross-dataset testing, demonstrating the robustness of SNNs across multiple datasets. However, limitations include novelty, insufficient baselines, and few evaluation datasets.  Overall, the writing is clear and technically thorough, effectively presenting both the potential and the limitations of SNNs in fake audio detection.

**Strengths:**

- **Comprehensive Hyperparameter Tuning**: The paper presents thorough hyperparameter tunings for SNN and CSNN models with various loss functions (e.g., CE-rate, CE-count) and surrogate gradients (e.g., Arctangent, Fast Sigmoid). This systematic tuning strengthens model performance.

- **Cross-Dataset Generalization**: Experiments evaluate model generalization across different datasets, testing models trained on Fake or Real against ASVspoof-2019 and vice versa.

- **Partial Fake Audio Detection**: The paper introduces a new partial fake audio dataset (PFA dataset) to explore model effectiveness in detecting partially manipulated audio at the frame level. This innovative frame-level classification demonstrates SNNs' potential for temporal precision in handling complex audio manipulations

**Weaknesses:**

- **Insufficient Baselines**: The methods compared in this study are relatively simple, lacking more advanced baseline methods that are widely recognized as state-of-the-art in synthetic speech detection. For a more comprehensive evaluation, SOTA methods like AASIST [1], AASIST-L [1], RawNet2 [2], and RawGATST [3] should be included, as they provide robust comparisons with current advanced approaches in audio anti-spoofing.

- **Few Evaluation Datasets**:  The experiments rely on only two datasets, ASVspoof-2019 and Fake or Real. While these are important datasets for this field, other available datasets, such as WaveFake [4] and In-the-Wild [5], would enhance the study’s scope by testing the model’s generalization across a broader array of synthetic audio sources, further verifying its performance on diverse data.

- **Lack of Robustness Evaluation**: The study does not explore the robustness of the SNN and CSNN models under common real-world audio perturbations, such as MP3 compression or Gaussian noise. Evaluating the models with these types of perturbations would provide a more practical understanding of their performance and adaptability in noisy or degraded audio environments.

### References
1. Jee-weon Jung, Hee-Soo Heo, Hemlata Tak, Hye-jin Shim, Joon Son Chung, Bong-Jin Lee, Ha-Jin Yu, and Nicholas Evans. AASIST: Audio anti-spoofing using integrated spectro-temporal graph attention networks. *ICASSP*, 2022.

2. Hemlata Tak, Jose Patino, Massimiliano Todisco, Andreas Nautsch, Nicholas Evans, and Anthony Larcher. End-to-end anti-spoofing with RawNet2. *ICASSP*, 2021.

3. Hemlata Tak, Jee-weon Jung, Jose Patino, Madhu Kamble, Massimiliano Todisco, and Nicholas Evans. End-to-end spectro-temporal graph attention networks for speaker verification anti-spoofing and speech deepfake detection. *ASVspoof 2021 Workshop*, 2021.

4. Joel Frank and Lea Schonherr. WaveFake: A dataset to facilitate audio deepfake detection. *NeurIPS Datasets and Benchmarks Track*, 2021.

5. Nicolas M. Muller, Pavel Czempin, Franziska Dieckmann, Adam Froghyar, and Konstantin Bottinger. Does audio deepfake detection generalize? *Interspeech*, 2022.

**Questions:**

- Would it be possible to conduct additional experiments incorporating these SOTA baselines to better contextualize the performance of the SNN and CSNN models? Given these models are considered state-of-the-art, including them could provide a stronger comparison.

-  Would the authors consider adding datasets like WaveFake or In-the-Wild, which feature different synthetic audio sources? Such datasets could better reveal the models’ generalization capability across various audio manipulation techniques.

- Since real-world audio is often subject to distortions such as compression and noise, how might the SNN and CSNN models perform under these common perturbations (e.g., MP3 compression, Gaussian noise)? Have the authors considered conducting robustness tests to simulate these conditions? Exploring this could add substantial practical relevance to the study, indicating the models’ adaptability in more variable environments.

- The SNN and CSNN models show parameter efficiency, but could the authors provide more details on their real-time applicability, especially in comparison to more complex models like AASIST? Given the reduced model size, what latency and computational gains are observed, and how might these impact deployment in resource-constrained settings?

---

> ### Author Response · Authors · 2024-11-19
>
> Thank you for providing a detailed evaluation of our work. To overcome the shortcomings mentioned in the review, we will make the following changes in the revised submission.
>
> 1. **Question**: Would it be possible to conduct additional experiments incorporating these SOTA baselines to better contextualize the performance of the SNN and CSNN models? Given these models are considered state-of-the-art, including them could provide a stronger comparison.
>
>     **Response**: We agree that comparing our proposed SNN and CSNN models against advanced methods would provide stronger contextualization and validation of our results. To address this, we will conduct additional experiments incorporating these SOTA baseline models into Experiment 2.
>
> 2. **Question**: Would the authors consider adding datasets like WaveFake or In-the-Wild, which feature different synthetic audio sources? Such datasets could better reveal the models’ generalization capability across various audio manipulation techniques.
>
>     **Response**: While WaveFake is an important dataset, we chose not to include it because it contains only fake samples, whereas our evaluation focuses on datasets that provide both fake and real samples, ensuring a balanced comparison for fake audio detection. To address the need for testing on more diverse and realistic audio manipulation scenarios, we will incorporate the In-the-Wild (Müller 2022) dataset into the cross-dataset evaluation experiment as part of the revision.
>
> 3. **Question**: Since real-world audio is often subject to distortions such as compression and noise, how might the SNN and CSNN models perform under these common perturbations (e.g., MP3 compression, Gaussian noise)? Have the authors considered conducting robustness tests to simulate these conditions? Exploring this could add substantial practical relevance to the study, indicating the models’ adaptability in more variable environments.
>
>     **Response**: We agree that exploring model performance in noisy or degraded audio environments would significantly enhance the practical relevance of our study. Previous studies (e.g., Patel et al., 2023) have demonstrated that SNNs exhibit greater robustness to input noise compared to ANNs. To build on these findings and provide a more comprehensive evaluation, we will conduct additional experiments to simulate real-world scenarios. Specifically, we will evaluate SNN and ANN performance under real-world conditions by testing with varying intensities of Gaussian noise and audio degraded by MP3 compression at different bitrates.
>
> 4. **Question**: The SNN and CSNN models show parameter efficiency, but could the authors provide more details on their real-time applicability, especially in comparison to more complex models like AASIST? Given the reduced model size, what latency and computational gains are observed, and how might these impact deployment in resource-constrained settings?
>
>     **Response**: As mentioned in the appendix E (Further Discussion), “A potential direction for future research is to compare the power efficiency of SNNs and traditional ANNs by implementing SNNs on neuromorphic hardware platforms, such as Intel Loihi (Davies et al., 2018)”. While neuromorphic hardware like Intel Loihi is specifically designed to leverage the efficiency of SNNs, the difficulty in obtaining such chips limits our ability to perform direct implementation and testing  of SNNs on these platforms. Comparing SNNs to ANNs solely on conventional hardware (CPUs, GPUs, and TPUs), which are optimized for ANN architectures, would lead to an unfair assessment of SNNs’ true latency and computational efficiency.
>
> **References**:
>
> Patel, K. P., & Schuman, C. D. (2023, April). Impact of noisy input on evolved spiking neural networks for neuromorphic systems. In Proceedings of the 2023 Annual Neuro-Inspired Computational Elements Conference (pp. 52-56).
>
> Müller, N. M., Czempin, P., Dieckmann, F., Froghyar, A., & Böttinger, K. (2022). Does audio deepfake detection generalize? Interspeech 2022, Incheon, Korea, 18-22 September 2022.
>
> Davies, Mike, et al. "Loihi: A neuromorphic manycore processor with on-chip learning." Ieee Micro 38.1 (2018): 82-99.

---

### Official Review · Reviewer_Ggm5 · 2024-11-05

**Soundness:** 1
**Presentation:** 2
**Contribution:** 1
**Rating:** 3
**Confidence:** 4

**Summary:**

This study explores the application of Spiking Neural Networks (SNNs) in detecting both fully fake and partially fake audio. It specifically evaluates and compares the performance of two proposed SNNs against several self-implemented Artificial Neural Networks (ANNs) and existing models, focusing on their ability to generalize across different datasets

**Strengths:**

This work clearly introduces the proposed method, including feature extraction and various SNN designs. I particularly appreciate how the Experiments section is organized with a clear presentation, making it very easy for the reviewer to follow.

**Weaknesses:**

1. The related work mentioned in this paper may not fully represent the current research status, as deep fake audio detection has been a long-standing and ongoing research direction. As stated in Section 1, "the detection of partially fake audio—where real and synthetic audio segments are seamlessly merged—remains an underexplored area", which may not be accurate. For example, research on identifying or localizing fake segments within a piece of audio can be considered as detecting partially fake audio, according to the taxonomy of this work. The following literature from 2022 to 2024 provides just a few examples that are not included in this paper:
 - Yadav, Amit Kumar Singh, et al. "Mdrt: Multi-domain synthetic speech localization." ICASSP 2024-2024 IEEE International Conference on Acoustics, Speech and Signal Processing (ICASSP). IEEE, 2024.
 - Xie, Yuankun, et al. "An Efficient Temporary Deepfake Location Approach Based Embeddings for Partially Spoofed Audio Detection." ICASSP 2024-2024 IEEE International Conference on Acoustics, Speech and Signal Processing (ICASSP). IEEE, 2024.
 - Zhang, Lin, et al. "Range-Based Equal Error Rate for Spoof Localization." arXiv preprint arXiv:2305.17739 (2023).
 - Khan, Awais, and Khalid Mahmood Malik. "Securing voice biometrics: One-shot learning approach for audio deepfake detection." 2023 IEEE International Workshop on Information Forensics and Security (WIFS). IEEE, 2023.

2. The motivation for introducing SNNs for detecting fake audio could have been more convincingly stated. For example, does the architecture of SNNs make them particularly suitable for this task? According to Section 1, "SNNs are inherently designed to process temporal data due to their ability to capture the timing and sequence of events, which makes them particularly suitable for tasks involving audio, which has rich temporal dynamics." However, the experimental results shown in Table 2 demonstrate that SNNs actually perform worse than the self-implemented CNN in most cases. Additionally, from an efficiency perspective, do SNNs offer other benefits beyond having fewer parameters that make them more suitable for practical use? If so, in what scenarios do existing continuous-parameterized neural networks fail to meet the detection latency requirements?

3. Since the reviewed related work is limited, this may also make the evaluation less comprehensive and representative. For example, as stated in Section 4.2, "Due to the lack of cross-dataset evaluation studies on assessing the generalization ability of ANNs for the fake audio detection problem...", this statement may not fully consider the necessity of cross-dataset generalization performance evaluation for existing deep neural network-based fake audio detectors. If the evaluation is limited to simple ANN architectures, such as feed-forward and convolutional architectures, then the considered evaluation scope could have been significantly expanded.

**Questions:**

N/A

---

> ### Author Response · Authors · 2024-11-19
>
> We appreciate your detailed feedback on our paper. To improve our work by answering your questions and remarks, we will make the following changes to the revised submission. Below are our responses to your numbered suggestions.
>
> 1. **Response**: We acknowledge that recent advancements in detecting partially fake audio, particularly those focusing on identifying or localizing fake segments, are not fully covered in the current version of the paper. To address this, we will thoroughly examine the references you provided, as well as other relevant recent works from 2022 to 2024, and incorporate their summaries into the related work section.
>
> 2. **Response**: While the experimental results in Table 2 indicate that SNNs perform worse than the self-implemented CNN in most cases, SNNs offer distinct advantages beyond accuracy. These include better cross-dataset Equal Error Rates (EERs), which highlight their robustness to dataset shifts, and computational efficiency due to sparse activation. Additionally, SNNs' inherent temporal processing capability makes them well-suited for tasks involving partial fake audio, as evidenced by their superior performance compared to artificial neural networks (ANNs) in Experiment 3. To address these points, we will enhance Section 6.2 to provide a more detailed explanation of the trade-offs between SNNs and other models, emphasizing the benefits of sparse activation and robustness. We will also include practical real-world scenarios, such as fake audio detection in call authentication systems, where latency and computational efficiency are critical requirements. These examples will illustrate how SNNs can be a viable alternative to ANNs in scenarios where continuous-parameterized neural networks fail to meet stringent detection latency or energy efficiency demands.
>
> 3. **Response**: We recognize that the statement in Section 4.2 regarding the lack of cross-dataset evaluation studies may not fully reflect the broader efforts in assessing generalization ability in existing work. To address this, we will expand our evaluation to include more recent and advanced ANN models, such as RawNet2 (Tak et al. 2021) and AASIST (Jung et al. 2022), which have demonstrated strong performance in fake audio detection tasks. Further, the cross-dataset evaluations of these models will help highlight the generalization challenges faced by ANNs and situate our SNN-based approach within a broader context.
> Additionally, we will update Section 2 (Related Work) to acknowledge and reference recent studies that have focused on cross-dataset generalization performance for deep neural network-based fake audio detectors.
>
> **References**:
>
> Tak, Hemlata, et al. "End-to-end anti-spoofing with rawnet2." ICASSP 2021-2021 IEEE International Conference on Acoustics, Speech and Signal Processing (ICASSP). IEEE, 2021.
>
> Jung, Jee-weon, et al. "Aasist: Audio anti-spoofing using integrated spectro-temporal graph attention networks." ICASSP 2022-2022 IEEE international conference on acoustics, speech and signal processing (ICASSP). IEEE, 2022.

---

> > ### Comment · Reviewer_Ggm5 · 2024-11-25
> > **Thanks the authors for their clarification**
> >
> > Thanks the authors for their clarification. After reviewing the authors' response and other reviewers' review, I would like to keep my original score.

---

### Official Review · Reviewer_SCyC · 2024-11-08

**Soundness:** 2
**Presentation:** 3
**Contribution:** 2
**Rating:** 5
**Confidence:** 4

**Summary:**

In response to the evolving need for fake audio detection, the authors proposed a Spiking Neural Network (SNN) based detection method for detecting fake and partially fake audio.

**Strengths:**

1. The idea is interesting, and the research motivation is clear.
2. Language is mostly accessible, though with some minor issues

**Weaknesses:**

My suggestion is as follows:
1. The author proposes in the contribution that "combining real and synthetic samples from the Fake or Real dataset, Demonstrating the effectiveness of SNNs in detecting partial fake audio at the frame level, "but no specific experimental data or demonstration examples were provided for the detection effect at the frame level. It is recommended that the author further improve this section.
2. The author points out in the introduction the limitations of existing ANN models in generalization and processing complex forged audio, but lacks data support or experimental evidence for the specific manifestations of these limitations. Suggest citing specific literature or experimental data to quantify the shortcomings of existing methods in detecting counterfeit audio.
3. The model proposed in the paper contains multiple key modules, but the role of these modules is not fully emphasized in the overall framework diagram or textual description. It is suggested that the author highlight the core role of each module in the methodology section to highlight the logical structure of the model and deepen readers' understanding of its working mechanism.
4. The author designed multiple experimental plans, but the organization of experimental design and results is somewhat scattered. To enhance logical coherence, suggest clearly separating the objectives, methods, and results of each experiment into sections.
5. The experimental results presented in Table 2, especially the EER index, perform worse than the existing methods, which the authors must explain in the paper.

**Questions:**

Please refer to the above suggestions

---

> ### Author Response · Authors · 2024-11-19
>
> Thank you for your detailed review. We aim to overcome these shortcomings by including the following upgrades in the revised version of the paper. Below are our responses to your numbered suggestions.
>
> 1. **Response** : Experiment 3 (Section 5.3) was designed to evaluate SNN’s ability to detect partial fake audio by classifying at the frame level. We will revise Section 5.3 to provide a more detailed explanation of the experiment's objective, methodology, and setup along with Section 6.3 (in results). This revision will clarify how frame-level classification was performed and how it ties into the evaluation of SNNs for partial fake audio detection.
>
>     Additionally, we will utilize the PartialSpoof (Zhang et al. 2022) dataset, which is more recent and uses advanced methods of blending fake and real samples to create partial fake samples, to train and test SNN models for partial fake audio detection.
>
> 2. **Response**: As mentioned in the Section 2 (Related Works), “Many SOTA models, while effective on the datasets they were trained on, struggle with generalizing to unseen TTS or VC models (Chen et al., 2020). The existing models in literature, typically trained on datasets containing entirely fake or entirely real samples, struggle to identify the manipulated portions when genuine audio is present (Rahman et al., 2022).”
>
>     To clearly demonstrate the limitations of existing ANN models in generalization and processing complex forged audio, we will include a table containing a summary of experimental results from previous papers.
>
> 3. **Response**: We agree that the current explanation in Section 4.3 (Spiking Neural Networks) is limited to high-level details and does not sufficiently clarify the contribution of individual modules.
> To address this, we will enhance Section 4.3 to provide a detailed breakdown of the fundamental roles of the core modules within the SNN framework, including Leaky, Convolution, MaxPool, and Fully-Connected layers.
>
> 4. **Response**: Section 5 (Experiments) contains three experiments to evaluate proposed models for fake and partial fake audio detection. We will upgrade each subsection of section 5 to clearly state the motivation and the objective of each experiment. Also, since each experiment contains multiple variants and multiple results, we will improve the writing in section 6 (Results) to differentiate and explain results of each sub-experiments more clearly.
>
> 5. **Response**: While the EER performance of SNN models is lower than that of existing methods, this is largely due to their reduced architectural complexity. However, SNNs offer other significant benefits, such as better cross-dataset generalization in EER performance, as they demonstrate robustness to dataset shifts. Moreover, their computational efficiency, stemming from sparse activation, makes them well-suited for resource-constrained scenarios.
> Additionally, SNNs leverage their temporal processing capabilities to outperform ANN models when evaluated on partial fake audio detection, as demonstrated in Experiment 3. These findings highlight the trade-offs between model complexity, accuracy, and computational efficiency, which are crucial for certain real-time applications.
> To address this in the paper, we will revise Section 6.2 to provide a clearer explanation of these trade-offs and the implications of the results presented in Table 2. This will include discussing the scenarios where SNNs are more advantageous, such as applications requiring low-latency detection with limited computational resources. These updates will provide a comprehensive explanation of the results and their significance.
>
> **References**:
>
> Chen, T., Kumar, A., Nagarsheth, P., Sivaraman, G., Khoury, E. (2020) Generalization of Audio Deepfake Detection. Proc. The Speaker and Language Recognition Workshop (Odyssey 2020).
>
> Rahman, Md Hafizur, et al. "Detecting synthetic speech manipulation in real audio recordings." 2022 IEEE International Workshop on Information Forensics and Security (WIFS). IEEE, 2022.
>
> Zhang, Lin, et al. "The partialspoof database and countermeasures for the detection of short fake speech segments embedded in an utterance." IEEE/ACM Transactions on Audio, Speech, and Language Processing 31 (2022): 813-825.

---

### Note · Authors · 2024-11-27

I have read and agree with the venue's withdrawal policy on behalf of myself and my co-authors.